# Segment Anything with Multiple Modalities

## Abstract

Robust and accurate segmentation of scenes has become one core functionality in various visual recognition and navigation tasks. This has inspired the recent development of Segment Anything Model (SAM), a foundation model for general mask segmentation. However, SAM is largely tailored for single-modal RGB images, limiting its applicability to multi-modal data captured with widely-adopted sensor suites, such as LiDAR plus RGB, depth plus RGB, thermal plus RGB, etc. We develop MM-SAM, an extension and expansion of SAM that supports cross-modal and multi-modal processing for robust and enhanced segmentation with different sensor suites. MM-SAM features two key designs, namely, unsupervised cross-modal transfer and weakly-supervised multi-modal fusion, enabling label-free and parameter-efficient adaptation toward various sensor modalities. It addresses three main challenges: 1) adaptation toward diverse non-RGB sensors for single-modal processing, 2) synergistic processing of multi-modal data via sensor fusion, and 3) mask-free training for different downstream tasks. Notably, we demonstrate that the output latent space of SAM's RGB image encoder can function as a highly abstract, shareable embedding space compatible with segmentation across different sensor modalities. Extensive experiments show that MM-SAM consistently outperforms SAM by large margins, demonstrating its effectiveness and robustness across various sensors and data modalities. Code will be released.

## 1 Introduction

Leveraging flexible geometric prompts with points, boxes, or coarse masks, the recent Segment Anything Model (SAM) (Kirillov et al., 2023) has emerged as a state-of-the-art visual foundation model for general mask segmentation. Despite its advanced capabilities, SAM's training on billions of RGB image masks has tailored it primarily for optical RGB cameras. Consequently, it often struggles or even fails when processing data from other visual sensor modalities.

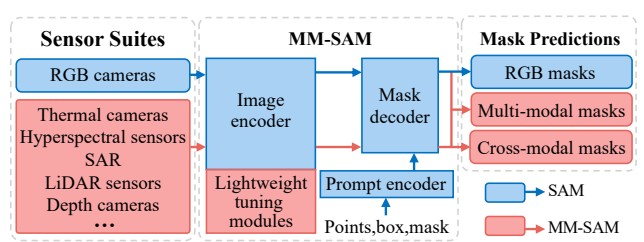

Figure 1: The proposed MM-SAM extends and expands SAM towards multi-modal data with various sensor suites, facilitating cross-modal and multi-modal segmentation without requiring mask annotations in different downstream tasks.

This limitation constrains the applicability of SAM, as we are facing increasing multi-modal data and sensor suites that integrate multiple sensors to capture complementary and paired data. It is crucial to extend SAM's capabilities beyond RGB cameras, enabling it to fully leverage the strengths of various sensor modalities. Such functional expansion of SAM can enhance its perception robustness and accuracy under complicated and dynamic situations.

This paper presents MM-SAM, a Multi-Modal SAM that extends and expands SAM toward multi-modal data captured with various sensor suites. Our goal, as illustrated in Figure 1, is to adapt pre-trained SAM with lightweight modules to enable cross-modal segmentation for individual sensor modalities and multi-modal segmentation with sensor fusion. To this end, MM-SAM addresses several major challenges while adapting SAM toward multi-modal data:

- *Adapting SAM for cross-sensor heterogeneous data.* We design *Unsupervised Cross-Modal Transfer* (UCMT) that incorporates modal-specific patch embedding module and parameter-efficient tuning

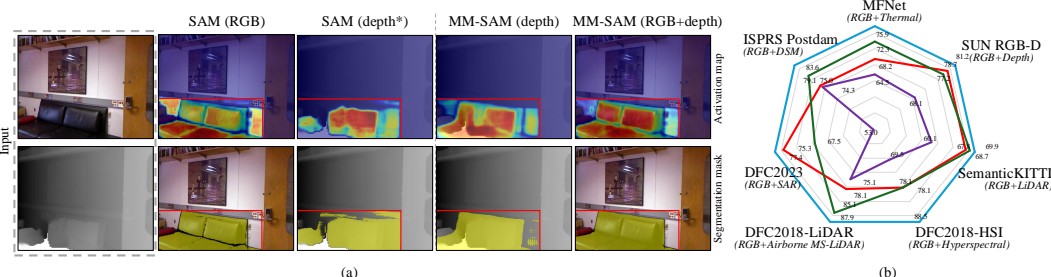

Figure 2: MM-SAM extends and expands SAM effectively. (a) Activation heatmap and mask predictions for segmenting the *sofa* in an example of RGB and depth images from SUN RGB-D (Song et al., 2015). With a box prompt (in red), MM-SAM performs clearly better for cross-modal segmentation of depth, and it also enables superb multi-modal segmentation with modality fusion. (b) MM-SAM demonstrate superior robustness and accuracy across seven multi-modal datasets, each featured by RGB plus a non-RGB X* modality. [●]: SAM on RGB, [●]: SAM on X*, [●]: MM-SAM on X with cross-modal adaptation, and [●]: MM-SAM on RGB+X with multi-modal fusion. The symbol * denotes false-color images[1] transformed from each non-RGB modality. The radius is normalized by MM-SAM's multi-modal segmentation scores. Bigger area coverage indicates better segmentation. Best viewed in color.

into SAM's image encoder, facilitating the extraction of modal-specific sensor features. UCMT includes an embedding unification loss that enforces unified representations across sensor modalities within the output latent space of SAM's image encoder, ensuring segmentation compatibility with the prompt encoder and mask decoder. This simple and lightweight design empowers MM-SAM with superior segmentation ability on individual modalities, as demonstrated in Figure 2.

- *Adapting SAM for synergistic sensor fusion*. We design *Weakly-supervised Multi-Modal Fusion* (WMMF), featuring a lightweight selective fusion gate for adaptive fusion of multi-modal embeddings. As illustrated in Figure 2, the selective fusion gate enables effective sensor fusion under complicated and dynamic situations, greatly enhancing segmentation robustness and accuracy compared to using individual modalities alone.

- *Label-free SAM adaptation towards different sensors*. MM-SAM requires no mask annotations for adaptation. Specifically, UCMT leverages unlabeled multi-modal data from sensor suites, while WMMF introduces multi-modal pseudo-labeling to train the selective fusion gate with given geometric prompts. The label-efficient adaptation expands MM-SAM's applicability significantly.

Through the straightforward design of MM-SAM, we demonstrate, for the first time, that the latent space output from SAM's RGB image encoder can serve as a highly abstract, shareable embedding space, compatible with segmentation across sensor modalities. By aligning embeddings from different sensor types within this unified space, MM-SAM enables efficient cross-modal segmentation and multi-modal fusion, effectively overcoming the inherent differences in sensor patterns and features.

Notably, MM-SAM's general framework is applicable to both the original SAM and the recently released SAM 2 (Ravi et al., 2024), demonstrating remarkable adaptability and effectiveness of the core idea across different architectures of the SAM models. This versatility highlights such framework as a powerful tool for future visual foundation model research in multi-modal tasks.

We highlight several key characteristics of MM-SAM. **Pioneering**: To the best of our knowledge, this is the *first* study to explore visual foundation models for sensor suites. **Simplicity**: The designed UCMT and WMMF are technically straightforward, enabling seamless integration with both SAM and SAM 2. **Efficiency**: MM-SAM achieves cost-efficient adaptation across multiple modalities. It introduces minimal trainable parameters and requires no manual annotations, making it highly effective for extending SAM's capabilities to various sensor types. **Robustness**: It demonstrates superior effectiveness across a broad spectrum of sensor modalities and diverse scene types.

---

[1]Non-RGB modality data are converted into false-color images with three channels to meet SAM's input requirements for zero-shot segmentation. See Appendix 6.2.1 for details.

## 2 RELATED WORKS

**Image Segmentation Foundation Model.** Scaling up deep neural networks has led to impressive advancements across various recognition tasks, inspiring the development of language and vision-language foundation models pre-trained on web-scale datasets (Brown et al., 2020; Radford et al., 2021), as well as vision foundation models such as SAM (Kirillov et al., 2023) and DINO v2 (Oquab et al., 2023). Among these foundation models, SAM is notable for its ability to perform zero-shot mask segmentation with flexible geometric prompts. Several studies explore adapting SAM to various specialized domains (Zhang et al., 2023c; Xiao et al., 2024) such as medical images (Ma et al., 2024), camouflaged objects (Chen et al., 2023), thin structures (Ke et al., 2024), and optical RGB remote sensing images (Zhang et al., 2024b; Chen et al., 2024). Additionally, some research expands SAM's capabilities beyond binary mask segmentation such as semantic recognition (Wang et al., 2023; Li et al., 2023; Zhang et al., 2023f; 2024a) and pose estimation (Lin et al., 2023). Efforts have also been made to enhance SAM towards more efficient and lightweight models (Zhao et al., 2023; Zhang et al., 2023a; Xiong et al., 2023).

On the other hand, SAM is constrained to RGB cameras due to its training on large-scale RGB image masks. Recent studies attempt to mitigate this limitation by transforming non-RGB data into false-color images to align with SAM's input requirements (Xiao et al., 2024; Gong et al., 2023) or re-training SAM with newly annotated data (Song et al., 2024; Li et al., 2024; Peng et al., 2023). However, data transformation can result in information loss and discrepancies with SAM's training distribution, leading to suboptimal segmentation performance. Re-training the model, on the other hand, is labor-intensive due to the significant effort in data collection and annotation. Given the prevalence of various sensor suites in perception tasks, it is crucial to extend SAM's capabilities to handle non-RGB and multi-modal data. MM-SAM is designed to fill this gap, enabling seamless integration of SAM with various sensor suites.

**Efficient Tuning** of foundation models has become more critical due to their growing size and high costs of deploying separate models for each task. Two primary approaches have been explored. The first is *parameter-efficient tuning*, such as Low-Rank Adaptation (LoRA) (Hu et al., 2021), prompt tuning (Zhou et al., 2022; Jia et al., 2022a), and adapters (Rebuffi et al., 2017; 2018; Gao et al., 2023; Xu et al., 2023), which work by freezing the core model and introducing a minimal number of learnable parameters. The second is *data-efficient tuning*, such as few-shot learning (Xiao et al., 2024) and weakly-supervised domain adaptation (Zhang et al., 2023b), which aims to achieve desired accuracy with minimal training data or annotations. While existing studies primarily focus on single-modal efficient tuning, the proposed MM-SAM aims to adapt for cross-modal and multi-modal processing while being parameter-efficient and label-efficient concurrently.

**Multi-Modal Fusion.** Fusing multi-modal data offers significant advantages by leveraging complementary information from different sources. However, this task is challenging due to data heterogeneity (Liang et al., 2023) and the need for complex calibration and alignment (Gupta et al., 2016). By incorporating non-RGB modalities such as depth (Cao et al., 2021; Hu et al., 2019), thermal (Zhang et al., 2021), LiDAR (Yan et al., 2022), etc., previous studies have demonstrated the benefits of multi-modal fusion in various visual detection and recognition tasks (Hazirbas et al., 2017; Zhuang et al., 2021; Hong et al., 2020; Zhang et al., 2023e;d). However, these methods rely on fully supervised learning with large scale annotated datasets and require tuning all model parameters, limiting their efficiency for visual foundation models that prioritize parameter-efficient tuning (Jia et al., 2022a) to preserve their powerful representations in low cost. MM-SAM addresses this by extending and expanding SAM to to sensor suites, enabling efficient fusion of multi-modal data without the need for ground-truth annotations. To the best of our knowledge, MM-SAM is the first framework that adapts SAM for sensor suites, significantly broadening its applicability across various downstream tasks.

## 3 METHODOLOGY

### 3.1 PRELIMINARIES

**Segment Anything Model.** SAM (Kirillov et al., 2023) consists of three key modules for image mask segmentation: a heavyweight *image encoder* (i.e., ViT (Dosovitskiy et al., 2020)) that encodes input images into image embeddings, a lightweight *prompt encoder* that encodes geometric prompts (such

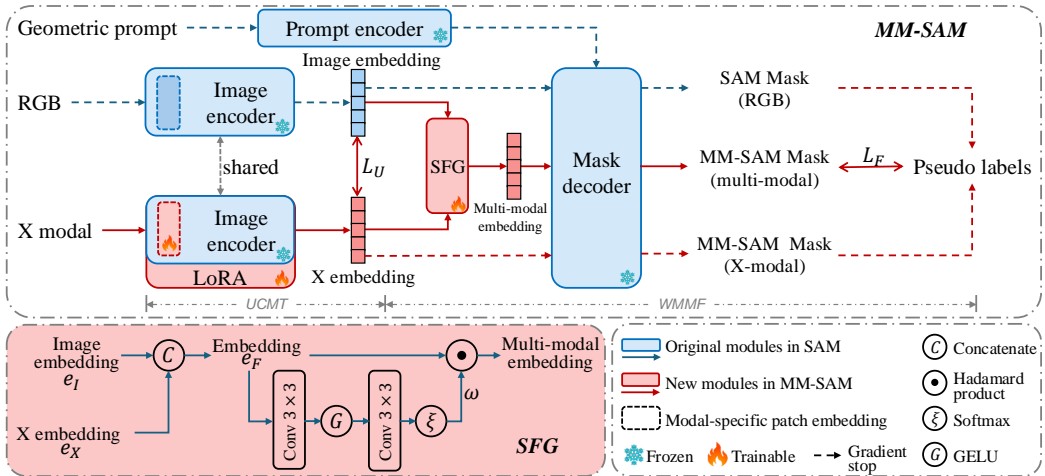

Figure 3: Overview of MM-SAM. MM-SAM freezes the entire SAM architecture while tuning it with multi-modal pairs (RGB and non-RGB modal X) for achieving cross-modal and multi-modal segmentation. It consists of two novel tuning modules: 1) Unsupervised Cross-Modal Transfer (UCMT) introduces modality-specific patch embedding module and low-rank (LoRA) structures into SAM's image encoder for extracting modality-specific X embeddings. An embedding unification loss ($L_U$) aligns X embeddings with SAM's RGB image embeddings to ensure segmentation compatibility; 2) Weakly-supervised Multi-Modal Fusion (WMMF) incorporates Selective Fusion Gate (SFG) to generate multi-modal embeddings, trained with multi-modal pseudo-labeling for adaptive sensor fusion. The whole training is mask-free. During inference, MM-SAM supports segmentation for single or multiple modality data.

as points, boxes, or coarse masks) into prompt embeddings, and a lightweight *mask decoder* that combines these embeddings to predict segmentation masks. SAM is trained on the SA-1B dataset, which includes over 11 million RGB images with 1.1 billion mask annotations. More details about SAM are described in (Kirillov et al., 2023). This work aims to extend and expand SAM toward cross-modal and multi-modal segmentation tasks, addressing the challenge of deploying SAM for various sensor suites.

**Sensor Suites with Modality Pairs.** A sensor suite is a collection of sensors deployed together within a system to capture data from different modalities for comprehensive sensing. This paper focuses on visual sensors such as RGB cameras and LiDAR scanners, widely used in visual recognition and navigation tasks. The multi-modal data captured by these sensors is naturally paired in space. We cover two main categories of sensor suites: 1) *Time-synchronized* suites, where multiple sensors are calibrated on a unified platform for simultaneous data collection; and 2) *Time-asynchronous* suites, where sensors are mounted on disparate platforms, capturing data at different times and perspectives but aligned through geographic coordinates. Representative examples include remote sensing sensors for earth observation. More details on the datasets are provided in Section 4.1.

## 3.2 MM-SAM

The main objective of MM-SAM design is to adapt SAM's image encoder to handle other modalities within SAM's segmentation pipeline. This requires the adapted image encoders to effectively encode modality-specific embeddings while maintaining segmentation compatibility, enabling seamless integration with SAM's prompt encoder and mask decoder for cross-modal segmentation. To this end, we directly align embeddings of non-RGB modalities with paired RGB embeddings, ensuring unified representations across sensor modalities within the latent space of SAM's image encoder.

This strategy offers three key advantages: 1) It only adapts the image encoder, leaving the prompt encoder and mask decoder unchanged, minimizing the addition of parameters to SAM's architecture. 2) It fully utilizes SAM's powerful image encoder pre-trained on billion-scale RGB masks since such extensive training data is nearly impossible to obtain for other modalities. 3) The unified embedding space across sensor modalities simplifies multi-modal fusion, as detailed in Section 3.2.2.

The overall pipeline of MM-SAM is depicted in Figure 3. Built upon the frozen SAM architecture, MM-SAM inherits SAM's powerful zero-shot segmentation capabilities for RGB images. Additionally, it introduces two key modules for parameter-efficient and label-efficient adaptation: **Unsupervised Cross-Modal Transfer** (UCMT) for cross-modal segmentation and **Weakly-supervised Multi-Modal Fusion** (WMMF) for multi-modal segmentation.

### 3.2.1 CROSS-MODAL SEGMENTATION WITH UCMT

As depicted in Figure 3, MM-SAM operates on pairs of modalities $(I, X)$ from sensor suites, where $I$ represents RGB images and $X$ denotes its corresponding observation in another modality. Similar to how SAM processes RGB images, $X$ is divided into fixed-sized patches with matching spatial resolutions. To process $X$ directly, we introduce a trainable patch embedding module at the beginning of the ViT architecture, adjusting input channel numbers to match $X$ while maintaining the output channel number consistent with SAM's original patch embedding module for RGB images. In addition, we introduce parameter-efficient tuning structures in the backbone to adaptively encode modal-specific features from $X$. Specifically, we use LoRA (Hu et al., 2021) in each transformer block of ViT for its efficiency and lightweight nature (see Section 4.3). More details are provided in Appendix 6.1.1.

Once $(I, X)$ are encoded into image and X embeddings $e_I$, $e_X$, UCMT optimizes the trainable parameters through unsupervised embedding unification:

$$L_U = ||e_I - e_X||_2^2. \tag{1}$$

Minimizing $L_U$ ensures that X-modal embeddings closely align with the established RGB embedding space from SAM's image encoder. This alignment ensures compatibility with SAM's prompt encoder and mask decoder, enabling seamless integration into SAM's segmentation pipeline. Despite its simplicity, this alignment approach achieves robust and superior adaptation across various non-RGB modalities, as detailed results in Section 4.

### 3.2.2 MULTI-MODAL SEGMENTATION WITH WMMF

WMMF, similar to UCMT, operates in the output embedding space of the image encoder and fuses data of multiple sensor modalities to generate more comprehensive embeddings. The core idea is to generate patch-wise weights conditioned on all input sensor modalities, enabling a weighted fusion of paired embeddings. This ensures robust sensor fusion and multi-modal segmentation that adapts to varying conditions. As illustrated in Figure 3, WMMF introduces two innovative designs to achieve multi-modality fusion, namely, Selective Fusion Gate (SFG) for multi-modal fusion and multi-modal pseudo-labeling for mask-free training.

**Selective Fusion Gate (SFG).** We concatenate embeddings $e_I$ and $e_X$ to form embedding $e_F$ and forward it to a weight filter comprising a two-layer convolution sub-network followed by a softmax layer. The outcome of the weight filter, i.e., the weights $\omega$, is applied to perform a patch-wise weighted average of the embedding $e_F$, producing the multi-modal embeddings $\hat{e}_F$, i.e.,

$$\hat{e}_F = \omega e_F = \omega_i e_{I_i} + (1 - \omega_i)e_{X_i}, \tag{2}$$

where $i$ denotes the patch index. Similar to $e_I$ and $e_X$, $\hat{e}_F$ can be integrated with SAM's prompt embeddings and jointly fed into the mask decoder for refined mask prediction $\hat{M}_F$. More details about the SFG structure are provided in Appendix 6.1.2.

**Multi-Modal Pseudo Labeling.** While supervised learning with human mask annotations is straightforward, it is costly and labor-intensive while handling many downstream applications. We design multi-modal pseudo-labeling to mitigate this issue. Given geometric prompts, MM-SAM generates two single-modal mask predictions $M_I$ and $M_X$ from data of RGB and X-modality, respectively. The predictions are then fused to produce a refined mask prediction $M_F$. Specifically, we derive $M_F$ by selecting the most confident predictions from corresponding patches of the paired modalities, and employ it as pseudo ground truth for SFG training:

$$L_F = L_{bce}(\hat{M}_F, M_F) + L_{dice}(\hat{M}_F, M_F), \tag{3}$$

where $L_{bce}$ denotes the binary cross-entropy loss and $L_{dice}$ represents the dice loss (Milletari et al., 2016).

Table 1: Comparison of trainable parameters between ViT-B (Dosovitskiy et al., 2020)-based SAM and MM-SAM with different sensor modality pairs (RGB+X). Channel numbers of individual X are indicated in brackets.

| Model | Trainable parameters | | | | | | |
|---|---|---|---|---|---|---|---|
| SAM | 91M | | | | | | |
| MM-SAM | X-Modal / Module | Thermal(1) | Depth(1) | LiDAR(4) | HSI(48) | MS-LiDAR(6) | SAR(1) DSM(1) |
| | UCMT | 344.8K | 344.8K | 934.7K | 9.6M | 1.3M | 344.8K 344.8K |
| | WMMF | 148.1K | 148.1K | 148.1K | 148.1K | 148.1K | 148.1K 148.1K |
| | Total | 492.9K | 492.9K | 1.1M | 9.7M | 1.5M | 492.9K 492.9K |

The whole tuning objective of MM-SAM is summarized as follows:

$$L = L_U + L_F. \tag{4}$$

**Expanding MM-SAM to Include More Sensor Modalities.** While our discussion has focused on two modalities $(I, X)$ for simplicity, the MM-SAM allows seamlessly integrating additional modalities by expanding SFG for generating fusion weights of more modalities. Incorporating more sensor types further enriches the segmentation system with a broader spectrum of information, leading to enhanced performance and versatility. Further experimental insights and discussions regarding the integration of additional modalities are provided in Section 4.2.2.

### 3.2.3 TRAINING AND INFERENCE

During training, we freeze the pre-trained SAM parameters and only update the newly-included trainable parameters in two phases. In the UCMT training phase, pairs of modalities are directly fed into the image encoder for optimization. In the WMMF training phase, parameters introduced in the previous stage remain frozen, while only the SFG is updated with provided geometric prompts.

During inference, MM-SAM supports segmentation for both single-modal and multi-modal data. For cross-modal segmentation, the encoded embedding $X$ from the image encoder is directly forwarded to the mask decoder alongside a geometric prompt for mask prediction, following SAM's process for RGB images. In multi-modal segmentation, the Selective Fusion Gate (SFG) integrates different modality embeddings to generate the final embeddings of the image encoder.

**Remark** (Efficiency). *The training of MM-SAM features two notable properties:*
- *Parameter Efficiency: Table 1 compares trainable parameters between SAM and MM-SAM across different data modalities (implemented with ViT-B (Dosovitskiy et al., 2020)), more details to be elaborated in Section 4. It is evident that MM-SAM introduces limited additional parameters yet enhances the performance significantly across diverse modalities.*
- *Label Efficiency: The entire tuning process of MM-SAM requires no mask annotations: UCMT operates in an unsupervised manner, using only unlabeled modality pairs, while WMMF is weakly-supervised with geometric prompts which are notably easier to collect than mask annotations.*

**Remark** (Insights). *To the best of our knowledge, this is the first study that explores visual foundation models for sensor suites. Our analysis of MM-SAM yields several key insights:*
- *MM-SAM proves the feasibility of sharing the output latent space of SAM's powerful image encoder across sensor modalities. This robust sharability allows capturing embeddings of different sensor data that are modality-specific yet still compatible with other modules in SAM (i.e., the prompt encoder and mask decoder), facilitating cross-modal segmentation.*
- *The shared latent space enables sensor fusion, wherein MM-SAM adaptively weights embeddings of different sensor modalities and generates more informative embeddings for enhanced segmentation.*
- *MM-SAM is a general framework and easily applicable to various sensor types, suggesting promising avenues for further research in visual foundation models and sensor fusion.*

## 4 EXPERIMENT

### 4.1 EXPERIMENT SETUP

We first describe the main experimental setups, with full details provided in the appendix.

Table 2: We benchmark MM-SAM across seven datasets with eight different sensor modalities.

| Sensor suite | Dataset | Modalities | Task | #Cls | #Train | #Test |
|---|---|---|---|---|---|---|
| Time-synchronized | MFNet | RGB, Thermal | Road Scene Seg. | 8 | 784 | 393 |
| | FreiburgThermal | RGB, Thermal | Road Scene Seg. | 12 | 20,853 | 64 |
| | SUN RGB-D | RGB, Depth | Indoor Scene Seg. | 37 | 5,285 | 5,050 |
| | SemanticKITTI | RGB, LiDAR | Road Scene Seg. | 8 | 19,130 | 4,071 |
| Time-asynchronized | DFC2018 | RGB, HSI, MS-LiDAR | Building Seg. | 1 | 12 | 2 |
| | DFC2023 | RGB, SAR | Building Seg. | 1 | 2,969 | 751 |
| | ISPRS Potsdam[2] | RGB, DSM | Building Seg. | 1 | 32 | 6 |

Table 3: Segmentation results on time-*synchronized* sensor suites using bounding box prompts. For MFNet, mIoU is reported for total/day/night, following the official criteria. The symbol * indicates false-color images transformed from each non-RGB modality.

(a) MFNet

| Model | Modal | mIoU |
|---|---|---|
| SAM | RGB | 68.2/72.6/65.1 |
| | Thermal* | 64.5/61.4/65.0 |
| MM-SAM | Thermal | 72.3/67.7/73.1 |
| | RGB+Thermal | **75.9/74.7/74.7** |

(b) SUN RGB-D

| Model | Modal | mIoU |
|---|---|---|
| SAM | RGB | 78.7 |
| | Depth* | 68.1 |
| MM-SAM | Depth | 77.2 |
| | RGB+Depth | **81.2** |

(c) SemanticKITTI

| Model | Modal | mIoU |
|---|---|---|
| SAM | RGB | 64.1 |
| | LiDAR* | 55.6 |
| MM-SAM | LiDAR | 65.1 |
| | RGB+LiDAR | **66.4** |

**Datasets.** We evaluated MM-SAM over two broad categories of sensor suites: *time-synchronized suites* and *time-asynchronous suites*, as described in Section 3.1. Table 2 summarizes the seven datasets, which cover a diverse range of non-RGB modalities including SUN RGB-D (Song et al., 2015) for *depth*, MFNet (Ha et al., 2017) for *thermal*, SemanticKITTI (Behley et al., 2019) for *LiDAR* in autonomous driving, DFC2018 (Prasad et al., 2020) for airborne *multispectral LiDAR* (MS-LiDAR) and *hyperspectral imaging* (HSI), DFC2023 (Sun, 2022) for *Synthetic Aperture Radar* (SAR), and ISPRS Postdam[2] for *Digital Surface Models* (DSM). Detailed descriptions of the seven datasets and their processing details are available in Appendix 6.2.

**Implementation Details.** Experiments were conducted on four NVIDIA A100 GPUs. Detailed hyperparameters used to tune each of the models reported in Tables 3, 4 are provided in Appendix 6.3.1.

## 4.2 SEGMENTATION RESULTS

### 4.2.1 TIME-SYNCHRONIZED SENSOR SUITES

Table 3 presents the segmentation performance of SAM and MM-SAM on time-synchronized sensor suites. SAM's performance is evaluated on RGB images. For reference, we also transform $X$ into false-color images (denoted as X*) and test it with SAM for comparisons. MM-SAM is evaluated on another paired modality data $X$ alone as well as RGB+$X$. Here, $X$ represents thermal images in MFNet, depth images in SUN RGB-D, and LiDAR point clouds in SemanticKITTI.

We can observe that SAM achieves much better segmentation on RGB images than on false-color images from other modalities due to distribution discrepancies. In contrast, MM-SAM improves segmentation consistently by large margins across three non-RGB modalities. Notably, in MFNet and SemanticKITTI, MM-SAM on thermal images and LiDAR point clouds even outperforms SAM on paired RGB images, highlighting potential limitations of RGB cameras and strengths of non-RGB sensors in different scenarios. In addition, MM-SAM demonstrates effective sensor fusion by consistently surpassing any individual modalities alone, underscoring its robustness and versatility across time-synchronized sensor suites. These results demonstrate the efficacy of MM-SAM in leveraging diverse sensor data with superior segmentation performance.

---

[2]ISPRS 2D Semantic Labeling Contest Potsdam (2016). Available from: `https://www.isprs.org/education/benchmarks/UrbanSemLab/2d-sem-label-potsdam.aspx`

Table 4: Segmentation results over time-*asynchronous* sensor suites using bounding box prompts. The symbol * denotes false-color images transformed from each non-RGB modality.

(a) DFC2023

| Model | Modal | IoU |
|---|---|---|
| SAM | RGB | 75.3 |
| | SAR* | 53.0 |
| MM-SAM | SAR | 67.5 |
| | RGB+SAR | **77.4** |

(c) ISPRS Potsdam

| Model | Modal | IoU |
|---|---|---|
| SAM | RGB | 75.0 |
| | DSM* | 74.3 |
| MM-SAM | DSM | 79.1 |
| | RGB+DSM | **83.6** |

(b) DFC2018

| Model | Modal | IoU |
|---|---|---|
| SAM | RGB | 78.1 |
| | HSI* | 69.5 |
| | MS-LiDAR* | 75.1 |
| MM-SAM | HSI | 78.1 |
| | MS-LiDAR | 85.1 |
| | RGB+HSI | 88.5 |
| | RGB+MS-LiDAR | 87.9 |
| | HSI+MS-LiDAR | 86.5 |
| | RGB+HSI+MS-LiDAR | **89.3** |

### 4.2.2 TIME-ASYNCHRONOUS SENSOR SUITES

We further evaluate MM-SAM over challenging time-asynchronous sensor suites. We examine it on commonly used earth-observation datasets that often involve significant time gaps and variations in scanning angles and resolutions, introducing substantial domain discrepancies across modalities. Table 4 presents experiments for RGB images paired with SAR in DFC2023, HSI and MS-LiDAR in DFC2018, and DSM in ISPRS Potsdam. Similar to the experiments in Table 3, MM-SAM demonstrates advanced cross-modal segmentation performance and harvests the benefits of multi-modal sensors effectively.

**MM-SAM for More Sensor Modalities.** Table 4 (b) examines MM-SAM's performance with suites containing three sensor modalities: RGB, HSI, and MS-LiDAR. MM-SAM achieves impressive cross-modal segmentation with both HSI and MS-LiDAR. Moreover, fusing RGB with either HSI or MS-LiDAR results in consistent segmentation improvements. Notably, combining all three modalities yields the best performance, surpassing both the fusion of any two modalities and the results from individual modalities. This highlights MM-SAM's scalability, showcasing its ability to accommodate additional sensors and develop more comprehensive perception systems for various applications.

**Fusion without RGB.** Another notable observation in Table 4 (b) is MM-SAM's ability to perform fusion of two non-RGB modalities $(X_1, X_2)$, without relying on paired RGB images (i.e., RGB+$X$). Specifically, by training on pairs of (RGB, HSI) and (RGB, MS-LiDAR), MM-SAM achieves effective fusion of HSI and MS-LiDAR ("HSI+MS-LiDAR" in the table). In the experiments, we first train two adapted image encoders for HSI and MS-LiDAR with Unsupervised Cross-Modal Transfer on (RGB, HSI) and (RGB, MS-LiDAR) pairs, respectively, without involving SFG. Then, we perform cross-modal segmentation on HSI and MS-LiDAR to generate multi-modal pseudo labels, which are used to train an SFG for (HSI, MS-LiDAR) fusion. The results show better segmentation than using HSI or MS-LiDAR alone, suggesting that MM-SAM can potentially be deployed in sensor suites without RGB cameras, revealing further opportunities for sensor fusion.

### 4.2.3 MM-SAM FOR SAM 2

Recently, Meta introduced SAM 2 (Ravi et al., 2024), which offers improvements in both accuracy and speed over the original SAM. Our proposed MM-SAM, incorporating the UCMT and WMMF modules, could be seamlessly integrated into the SAM 2 framework. We extend MM-SAM to this updated version, referring to it as "MM-SAM 2". For further details of model structures, please refer to Appendix 6.5.

We conducted extensive experiments across various sensor suites to compare SAM 2 and MM-SAM 2, with results summarized in Table 5. Compared to the results in Tables 3 and 4, SAM 2 delivers notably better zero-shot performance on RGB images and most other sensor modalities, confirming its improvements over SAM. Significantly, MM-SAM 2 consistently surpasses SAM 2 in cross-modal and multi-modal segmentation across diverse sensor suites, aligning with MM-SAM's performance advantage over SAM. These results validate our key insight that SAM's RGB image encoder produces a highly abstract and shareable latent space, suitable for segmentation across different sensor modalities, further highlighting the robustness and versatility of MM-SAM's design.

Table 5: Segmentation results of MM-SAM 2 across various sensor suites. "X" represents non-RGB modalities; "SKT" denotes SemanticKITTI dataset. The symbol * denotes false-color images transformed from each non-RGB modality.

| Model (X) | SUN RGB-D (Depth) | MFNet (Thermal) | SKT (LiDAR) | DFC18 (HSI) | DFC18 (MS-LiDAR) | DFC23 (SAR) | Postdam (DSM) |
|---|---|---|---|---|---|---|---|
| SAM 2 (RGB) | 82.9 | 74.0/76.6/71.8 | 67.0 | 84.3 | 84.3 | 79.2 | 85.5 |
| SAM 2 (X*) | 71.9 | 70.5/65.4/71.9 | 53.7 | 75.3 | 83.9 | 58.1 | 75.9 |
| MM-SAM 2 (X) | 80.6 | 76.4/69.3/78.1 | 67.7 | 82.8 | 87.6 | 64.9 | 82.2 |
| MM-SAM 2 (RGB+X) | 84.2 | 78.5/76.8/78.0 | 68.6 | 90.3 | 91.0 | 79.4 | 87.3 |

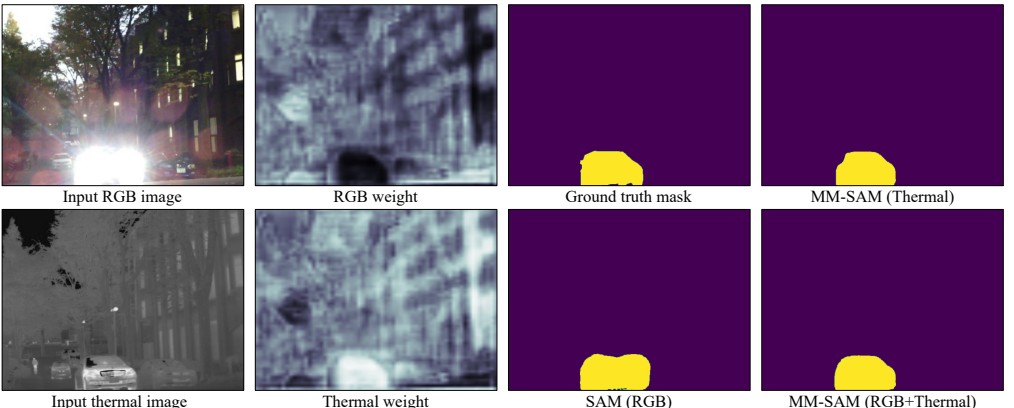

Figure 4: Visual illustration of adaptive fusion for enhanced segmentation with MM-SAM, using one sample of paired RGB and thermal images from the MFNet dataset. The second column shows fusion weights from the SFG, where brighter areas represent higher weights.

## 4.3 DISCUSSIONS AND ANALYSIS

**Visual Analysis of Selective Fusion Gate (SFG).** To understand how SFG dynamically adjusts the weighting of different sensors based on multi-modal inputs, we analyze an example from the MFNet dataset, as shown in Figure 4. In this scenario, the car's high beam creates strong light interference, making it challenging to recognize and segment the car in the RGB image. In contrast, the thermal image remains unaffected by the visible light. Consequently, SFG assigns a significantly higher weight to the thermal image in this area and a much lower weight to the corresponding RGB image area. This adaptive weighting results in more accurate segmentation. This example demonstrates how SFG manages complex and dynamic situations within sensor suites, effectively leveraging the strengths of each modality to improve segmentation robustness and accuracy.

**Zero-shot Segmentation.** We evaluated the generalization ability of MM-SAM on unseen domains for zero-shot segmentation tasks MFNet→FreiburgThermal (Vertens et al., 2020) datasets (both with RGB plus thermal) and SUN RGB-D→NYU (Nathan Silberman & Fergus, 2012)&B3DO (Janoch et al., 2013) datasets (all with RGB plus depth). As detailed in Appendix 6.3.2, for MFNet→FreiburgThermal, we use the model trained on MFNet (in Table 3 (a)) and test it on FreiburgThermal; While for SUN RGB-D→NYU&B3DO, we re-trained MM-SAM using the SUN RGB-D training set but excluding its subsets NYU&B3DO for cross-sensor testing (Song et al., 2015). The results are presented in Table 6. MM-SAM demonstrates superior and consistent zero-shot segmentation performance for both cross-modal segmentation and multi-modal fusion. The trend mirrors the positive results observed in previous intra-domain evaluations. These findings underscore the zero-shot potential of MM-SAM, highlighting its generalizability and effectiveness in segmentation to unseen domains.

**MM-SAM with different tuning approaches.** We assessed the effectiveness of various parameter-efficient tuning (PEFT) methods within MM-SAM for extracting modality-specific features. Specifically, we integrated three state-of-the-art PEFT methods: LoRA (Hu et al., 2021), Adapter-

Table 6: Zero-shot Segmentation results. For FreiburgThermal (Vertens et al., 2020), mIoU is reported for total/day/night. The symbol * denotes false-color images transformed from each non-RGB modality.

(a) MFNet→FreiburgThermal

| Model | Modal | FreiburgThermal |
|---|---|---|
| SAM | RGB | 65.3/71.8/60.7 |
| | Thermal* | 62.2/61.9/62.3 |
| MM-SAM | Thermal | 66.5/66.2/66.4 |
| | RGB+Thermal | **70.8/72.8/69.0** |

(b) SUN RGB-D→NYU&B3DO

| Model | Modal | NYU | B3DO |
|---|---|---|---|
| SAM | RGB | 79.5 | 77.4 |
| | Depth* | 69.0 | 67.1 |
| MM-SAM | Depth | 75.5 | 73.4 |
| | RGB+Depth | **81.4** | **80.1** |

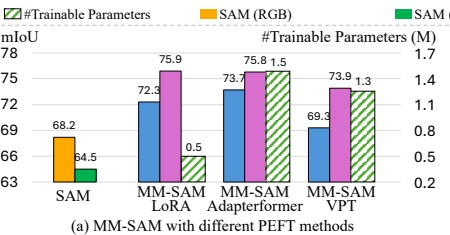
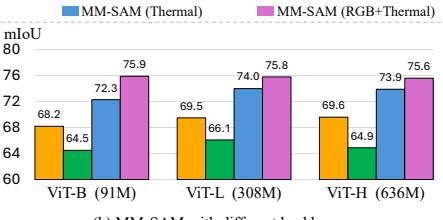

Figure 5: Segmentation performance of MM-SAM on the MFNet ("Total" split) using different parameter-efficient tuning (PEFT) methods in (a) and various ViT backbones in (b).

Former (Chen et al., 2022), and VPT (Jia et al., 2022b) in the image encoder. Figure 5 (a) compares the number of introduced trainable parameters and their performance on the MFNet dataset. The results show that LoRA introduces the fewest trainable parameters while achieving the best performance. We thus empirically select LoRA for the final implementation of MM-SAM. Nevertheless, all three PEFT methods demonstrate improved cross-modal segmentation and exceptional multi-modal segmentation capabilities, indicating MM-SAM's versatility and compatibility with various PEFT methods.

**MM-SAM with different backbones.** We evaluate MM-SAM's performance using various backbones for SAM's image encoder. Figure 5 (b) presents results of MM-SAM variants with ViT-B, ViT-L, and ViT-H (Dosovitskiy et al., 2020) based image encoders on the MFNet dataset, following the same setup as in Table 3 (a). The results show that MM-SAM is robust to backbone variations and achieves consistently advanced cross-modal and multi-modal segmentation across different encoder architectures.

**Limitations.** Though MM-SAM just introduces a small number of extra parameters, it remains computationally intensive and cannot operate at real-time speeds because of its dependence on SAM. This reliance demands substantial GPU resources, restricting its use in applications like video processing. In addition, similar to SAM, it is limited to binary mask segmentation and does not perform semantic or panoptic segmentation. Training MM-SAM requires paired modalities with RGB images, meaning an RGB camera must be included in sensor suites to collect training data. However, this constraint does not apply during inference.

## 5 CONCLUSION

In this study, we extended and expanded the Segment Anything Model (SAM) to accommodate various sensor suites. We proposed MM-SAM, a parameter-efficient and label-efficient adaptation method that enhances SAM's capabilities for cross-modal and multi-modal segmentation. By utilizing mask-free training, our approach significantly improves adaptation efficiency. Extensive evaluations across seven datasets and eight different sensor modalities demonstrated that our method significantly enhances SAM's robustness and performance in complex and dynamic scenarios. We hope that MM-SAM can lay a strong foundation and encourage future research to provide deeper insights into visual foundation models for sensor suites.

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

# 6 APPENDIX

## 6.1 MM-SAM DETAILED STRUCTURE

### 6.1.1 IMAGE ENCODER FOR NON-RGB MODALITIES

**Modality-specific path embedding.** To process non-RGB modality data, we introduce a new patch embedding module at the beginning of SAM's image encoder backbone (i.e., ViT (Dosovitskiy et al., 2020)). This module generates modality-specific patches. The input to the new patch embedding is $(B, D, 1024, 1024)$ compared to the original $(B, 3, 1024, 1024)$, both producing the same output sizes. Here, $B$ is the batch size, and $D$ represents the dimension of the specific modality, such as 1 for depth images, as detailed in Appendix 6.2.1.

**LoRA (Hu et al., 2021) for parameter-efficient tuning.** To learn modality-specific features, we integrate LoRA structures into each transformer block of SAM's pre-trained encoder. Each LoRA block uses a rank of 4, balancing the learning of modality-specific features with the number of tuning parameters. More descriptions can be found in (Hu et al., 2021).

### 6.1.2 SELECTIVE FUSION GATE

As shown in Figure 3, when fusing $K$ modalities, we first generate embeddings from the image encoder, denoted as $e_{X_k}$ for $k = 1, \ldots, K$. The RGB embedding can be one of them, denoted as $e_{X_j}$ (i.e., $e_{X_j} = e_I$). All embeddings have the same size $(B, 256, 64, 64)$. We concatenate these embeddings into $e_F$ with a size of $(B, 256 \times K, 64, 64)$, which is then input to the weight module. This module consists of two convolutional layers with $3 \times 3$ kernels, a GELU activation function, and a softmax layer. The convolutional layers have output channels of $16 \times K$ and $1 \times K$, respectively. The final softmax output, denoted as $\omega$ with a size of $(B, K, 64, 64)$, performs a Hadamard product with $e_F$ as described in Equation 2.

## 6.2 DATASETS AND METRICS

We conduct the experiments on time-synchronized and time-asynchronized sensor suites: 1) *Time-synchronized*, including MFNet (Ha et al., 2017) (RGB-Thermal), SUN RGB-D (Song et al., 2015) (RGB-Depth) and SemanticKITTI (Behley et al., 2019) (RGB-LiDAR); 2) *Time-asynchronized*, including Data Fusion Contest 2018 Dataset (DFC2018) (Prasad et al., 2020) (RGB-Hyperspectral-Multispectral LiDAR), Data Fusion Contest 2023 Dataset (DFC2023) (Sun, 2022) (RGB-SAR) and ISPRS Potsdam Dataset[2] (RGB-DSM).

**MFNet** (Ha et al., 2017) is a multi-sepctral RGB-thermal image dataset for autonomous driving research. Collected with an InfRec R500 camera, the dataset offers 1,569 densely annotated and synchronized RGB and thermal images across eight common driving obstacles captured in both day and night conditions. It provides eight classes of pixel-wise annotations for semantic segmentation. We use the original data split as described in the original paper (Ha et al., 2017). In this work, we use all eight classes, including car, person, bike, curve, car stop, guardrail, color cone and bump for per-class Intersection-over-Union (IoU) and report their mean Intersection-over-Union (mIoU) of all classes. Since no samples containing 'guardrail' category in daytime data, we report mIoU of daytime ("day" in Table 3 (a)) using only the rest seven classes. For "total" and "night" in the table, mIoU is calculated on all eight classes. More details of this dataset can be found at https://www.mi.t.u-tokyo.ac.jp/static/projects/mil_multispectral/.

**FreiburgThermal** (Vertens et al., 2020) is a dataset consisting of over 20,000 synchronized RGB and thermal images collected across different urban and rural environments during both day and night. It features pixel-wise semantic annotations of 12 classes. The dataset is designed to enhance research in thermal image segmentation. FreiburgThermal is valuable for multi-modal semantic segmentation tasks, especially in varying lighting conditions. We adopt the original dataset split. In this work, we use 12 classes for evaluation, including road, sidewalk, building, curb, fence, pole, vegetation, terrain, sky, person, car and bicycle, and report their mIoU. More details of this dataset can be found at http://thermal.cs.uni-freiburg.de.

**SUN RGB-D** (Song et al., 2015) is an RGB-Depth dataset for visual scene understanding. It includes 10,335 RGB and depth images of indoor environments captured by different types of

RGB-D cameras, with the RGB and depth images precisely aligned at the pixel level to enable accurate data fusion and analysis. Each image is annotated with detailed semantic segmentation labels of 37 categories. We follow the official data split for experiments. In this work, we use all 37 classes for evaluation and report their mIoU. More details of this dataset can be found at https://rgbd.cs.princeton.edu.

**SemanticKITTI** (Behley et al., 2019) is a outdoor point cloud dataset designed for 3D semantic segmentation within the context of autonomous driving. Every scene in this dataset is captured using a Velodyne-HDLE64 LiDAR sensor. It includes 22 sequences, which are split into different subsets: a training set comprising 10 sequences with 19,130 frames, a validation set that includes 1 sequence with 4,071 frames, and a testing set containing 11 sequences with 20,351 frames. Point-wise annotations of 32 classes are provided. We follow the original data split used in the SemanticKITTI dataset. In this work, we use 8 foreground classes with mask annotations for evaluation and report their mIoU. More details of this dataset can be found at http://semantic-kitti.org.

**DFC2018** (Prasad et al., 2020) contains 14 tiles of multi-source optical imagery from Houston, Texas. It features co-registered Very High Resolution (VHR) color images, hyperspectral images, and multispectral LiDAR point clouds. Hyperspectral data covers 380-1050 nm spectral range with 48 bands while multispectral LiDAR provides point cloud data at 1550 nm, 1064 nm, and 532 nm with intensity rasters from first return per channel. The dataset covers $4172 \times 1202m^2$ square meters with spatial resolution $5cm/pixel$ (0.05m GSD) for RGB images, $100cm/pixel$ (1m GSD) for HSI images, and $50cm/pixel$ (0.5m GSD) for MS-LiDAR. To test our fine-grained segmentation ability, we relabeled two tiles (272652_3289689, 273248_3289689) from the test set with super high quality building masks serving as evaluation ground-truth, which will be released together with code. The dataset is used for *building* segmentation in this paper. More details of this dataset can be found at https://hyperspectral.ee.uh.edu/?page_id=1075.

**DFC2023** (Sun, 2022) focuses on building detection using high-resolution optical satellite imagery and Synthetic Aperture Radar (SAR) images. The dataset encompasses buildings from 17 cities across 6 continents. We use this dataset to segment *buildings*. Specifically, data from Soochow and Copenhagen are used as the test set, while data from the remaining cities constitutes the training set. More details of this dataset can be found at here.

**ISPRS Potsdam** contains 38 high-resolution images of Potsdam City, Germany, with a ground sampling distance of 5 cm. This dataset includes two modalities: true orthophoto (TOP) and digital surface model (DSM). The TOP modality corresponds to RGB images, while the DSM modality includes the near-infrared band. In this study, we utilize both TOP and DSM data to construct a cross-modal and multi-modal learning paradigm. We designate images 6_07, 6_08, 6_09, 7_07, 7_08, and 7_09 as the test set, using the remaining images for training. In this paper, we ulilize this dataset for *building* segmentation and report IoU performance. More details of this dataset can be found at https://www.isprs.org/education/benchmarks/UrbanSemLab/2d-sem-label-potsdam.aspx.

### 6.2.1 DATA REPRESENTATIONS

**MM-SAM.** We use the standard RGB representations for **visual** images. For **LiDAR** point clouds from SemanticKITTI, we follow common practices in projection-based methods (Wu et al., 2018; Jaritz et al., 2020; Xiao et al., 2021) and project them into 4-channel images with coordinates $(x, y, z)$ and laser reflectance intensity. We use a single-channel image for the **thermal** data from MFNet and FreiburgThermal datasets due to its natural form in which current infrared thermal sensors return data. For **depth** images from SUN RGB-D, we use the single channel of depth. In DFC2018, for **hyperspectral** imaging, we directly use the original 48 channels with full information; While for **multispectral-LiDAR** data, we use officially provided LiDAR point cloud tiles to project $x, y, z$ onto rasters and generate 6 channel data, including geo-coordinates and intensity rasters at wavelengths of C1 (1550 nm), C2 (1064 nm), and C3 (532 nm). For the **SAR** images from DFC2023, we follow the official data format and use single-channel images. For the **digital surface model** data from Potsdam, we also use the single-channel (height value) images as input.

**SAM for false-color images from non-RGB modalities.** To meet the input requirements for SAM's processing, we convert all non-RGB modalities into three-channel false-color images. We use typical false colorization on single-channel images, i.e., normalizing them before stacking them into three

Table 7: Training hyperparameters of MM-SAM including UCMF and WMMF.

|  | UCMT | WMMF |
|---|---|---|
| Total epochs | 50 | 30 |
| Batch size | 16 | 16 |
| Optimizer | AdamW | AdamW |
| Peak learning rate | `1.6e-3` | `4e-4` |
| Scheduler | CosineAnnealingLR | CosineAnnealingLR |
| Minimum learning rate | eta_min=1e-5 | eta_min=1e-5 |
| Input prompts | - | Boxes |

Table 8: Data augmentation strategies over different datasets.

| | |
|---|---|
| SUN RGB-D | z-score, RandomCrop with a scale factor of [0.8, 1.0] |
| MFNet | z-score, RandomCrop with a scale factor of [0.8, 1.0] |
| DFC2023 | z-score, RandomCrop with a scale factor of [0.8, 1.0] |
| Postdam | log+min-max, RandomCrop with a scale factor of [0.8, 1.0] |
| DFC2018 | z-score |
| SemanticKITT | - |

channels. We apply this false colorization on thermal images and SAR images. For depth data, we follow common practice and map depth values to disparity before false colorization. Point clouds from SemanticKITTI are converted to depth data and conducted with false colorization. For hyperspectral imaging, we use the default bands of RGB channels. For multispectral-LiDAR data, we directly stack C1, C2, and C3 bands. For the DSM model from the Potsdam dataset, we perform a log normalization process to standardize the elevation values, followed by generating false colorization similar to depth.

## 6.3 IMPLEMENTATION DETAILS

### 6.3.1 TRAINING IMPLEMENTATION DETAILS

Table 7 provides the hyperparameters used to train each model reported in Tables 3 and 4. Table 8 lists the data augmentations applied for UCMT on each dataset, while no augmentations are used for WMMF. MM-SAM for all tested datasets could be trained with 4 NVIDIA A100 GPUs within 20 hours except for SemanticKITTI which took 35 hours. Note that we adopted simple training configurations for MM-SAM across different benchmarks. While more sophisticated tuning could potentially improve performance, it is not the main objective of our study.

Table 9: Segmentation performance on MFNet by MM-SAM with different embedding unification losses. mIoU is reported for total/day/night.

| Loss Type | Thermal | RGB+Thermal |
|---|---|---|
| $L_1$ loss | 71.7/66.7/72.4 | 75.5/74.8/74.1 |
| $L_2$ loss | 72.3/67.7/73.1 | 75.9/74.7/74.7 |
| Cosine similarity loss | 72.6/67.6/73.2 | 75.5/74.8/74.2 |

### 6.3.2 ZERO-SHOT EXPERIMENTAL DETAILS

**MFNet→FreiburgThermal.** For the zero-shot results in Table 6 (a), we follow the same strategy as for intra-domain evaluation, using the official training set of MFNet and the testing set of FreiburgThermal.

**SUN RGB-D→NYU&B3DO.** SUN RGB-D consists of data collected from four types of sensors: Kinect v1, Kinect v2, Xtion, and Realsense. For testing, we use data from Kinect v1 (specifically its NYU and B3DO subsets), while the remaining sensors are used for training, creating a robust cross-sensor evaluation of zero-shot segmentation as in Table 6 (b).

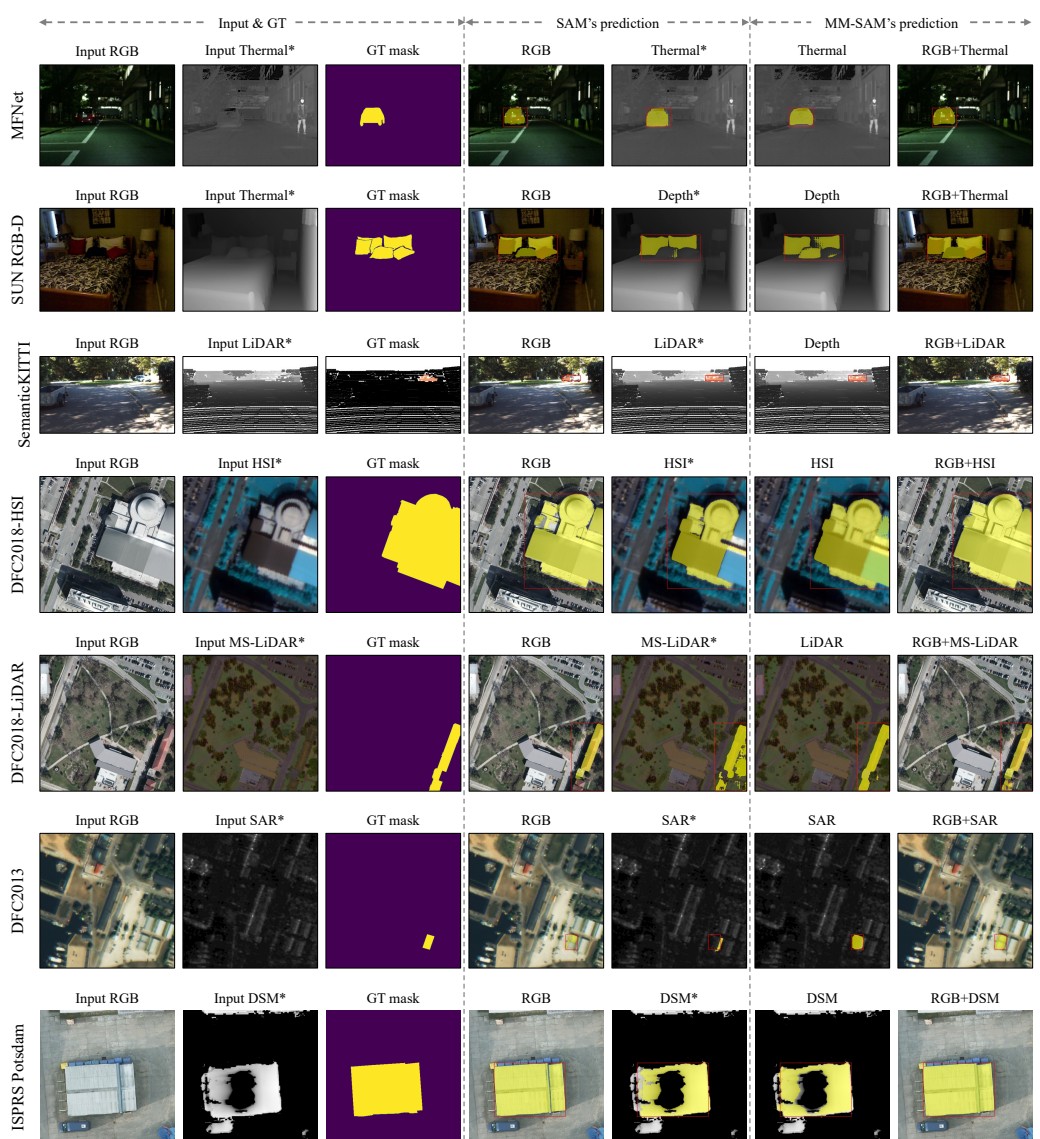

Figure 6: Visual comparisons of SAM (Kirillov et al., 2023) and MM-SAM. Red boxes denote geometric prompts, colored regions are mask predictions. The symbol * denotes false-color images transformed from each non-RGB modality.

## 6.4 ADDITIONAL RESULTS

**Design Choices in Losses.** We tested different losses for embedding unification in UCMT, including $L_1$ loss, $L_2$ loss, and Cosine Similarity loss. Table 9 shows segmentation results of MM-SAM trained with these different losses on MFNet, including cross-modal segmentation on thermal and multi-modal segmentation on RGB+thermal. All three losses achieved superior results, with the $L_2$ loss showing the best multi-modal segmentation result, though with only a marginal gap from the other two. We thus empirically select $L_2$ in our implementation. The results demonstrate that MM-SAM is robust to different losses.

**Visual Illustrations.** Figure 6 shows qualitative comparisons between SAM and MM-SAM across multiple segmentation tasks. These illustrations demonstrate how our proposed MM-SAM achieves superior cross-modal and multi-modal segmentation.

Table 10: Comparison of state-of-the-art semantic segmentation and multimodal fusion methods (upper part) with SAM and MM-SAM (lower part).

(a) SUN RGB-D

| Model | mIoU |
|---|---|
| CMX (Zhang et al., 2023d) | 52.4 |
| DFormer-L (Wang et al., 2022a) | 52.5 |
| DPLNet (Dong et al., 2023) | 52.8 |
| TokenFusion (Wang et al., 2022b) | 53.0 |
| GeminiFusion (Jia et al., 2024) | 54.6 |
| SAM (RGB) | 78.7 |
| SAM (Depth*) | 68.1 |
| MM-SAM (Depth) | 77.2 |
| MM-SAM (RGB+Depth) | 81.2 |

(b) MFNet

| Model | mIoU |
|---|---|
| DPLNet (Dong et al., 2023) | 59.3 |
| CMX (Zhang et al., 2023d) | 59.7 |
| CMNeXt (Zhang et al., 2023e) | 59.9 |
| Sigma-base (Wan et al., 2024) | 61.3 |
| CRM_RGBTSeg (Shin et al., 2023) | 61.4 |
| SAM (RGB) | 71.5 |
| SAM (Thermal*) | 68.2 |
| MM-SAM (Thermal) | 75.2 |
| MM-SAM (RGB+Thermal) | 78.4 |

**More Comparisons.** We benchmark MM-SAM and SAM against state-of-the-art (SOTA) segmentation and multimodal learning methods on the SUN RGB-D[1] and MFNet datasets[2]. Specifically, the SOTA methods primarily focus on semantic segmentation, SAM and MM-SAM are designed for prompted mask segmentation. Moreover, the SOTA approaches rely on fully supervised learning, while SAM operates in a zero-shot setting, and MM-SAM employs mask-free training. Despite these differences, the comparison offers valuable insight into how visual foundation models like SAM and MM-SAM perform in similar tasks.

The results are shown in Table 10. To ensure a fair comparison in metric numbers, we re-evaluated MM-SAM on MFNet by including the 'unlabeled' class to align with the standard evaluation criteria used in SOTA methods. We can see that SAM, even testing in a zero-shot setting, demonstrates powerful segmentation abilities as a foundation model, surpassing SOTA methods on both datasets by large margins. Moreover, the proposed MM-SAM achieves significantly better results than all of the compared methods, further validating its superiority in processing cross-modality and multi-modality data.

## 6.5   MM-SAM 2

The recently released SAM 2 (Ravi et al., 2024) extends SAM to both video and image domains with better accuracy using fewer interactions. Alongside the image encoder, prompt encoder, and mask decoder from SAM, SAM 2 introduces a memory mechanism, including memory attention, a memory encoder, and a memory bank for handling consecutive video frames. For more details, please refer to (Ravi et al., 2024).

The core designs of MM-SAM, including UCMT and WMMF, could also integrate seamlessly with SAM 2, allowing for an extension of its capabilities to sensor suites. We denote this version as "MM-SAM 2". Like with MM-SAM for SAM, we incorporate LoRA structures into the image encoder, i.e., ViT (Dosovitskiy et al., 2020), and introduce UCMT and WMMF with minor adjustments tailored to SAM 2.

- **UCMT:** Unlike SAM, SAM 2 uses multi-scale features from its image encoder. Specifically, it utilizes lateral features from stages 1 and 2 and the final vision features from the last two stages of the ViT transformer for mask decoding. Consequently, in addition to aligning vision features as in Equation 1, we also align lateral features from the corresponding stages between RGB and non-RGB data pairs.
- **WMMF:** Like MM-SAM, WMMF in MM-SAM 2 fuses features from both modalities through Equation 2. However, while MM-SAM only fuses the image and X features of the final stage of the image encoder, MM-SAM 2 performs separate fusions at different levels as its UCMT module: the lateral features from stages 1, 2, and the features from the last two stages of the ViT transformer.

---

[1]Benchmark results for open-sourced methods were retrieved from https://paperswithcode.com/sota/semantic-segmentation-on-sun-rgbd. Accessed on Sep. 29, 2024.

[2]Benchmark results for open-sourced methods were retrieved from https://paperswithcode.com/sota/thermal-image-segmentation-on-mfn-dataset. Accessed on Sep. 29, 2024.

Each fusion is handled by a specific SFG customized with corresponding input channels, enabling a hierarchical fusion mechanism specifically tailored for SAM 2, enhancing its capacity to process multi-modal data more effectively across different feature layers.

## 6.6 BROAD IMPACT

MM-SAM is environmentally friendly due to its resource-efficient design, including both parameter and label efficiency. It enhances the robustness of perception systems, particularly in challenging and dynamic conditions, by integrating various sensors. Additionally, MM-SAM improves AI-assisted labeling in areas where SAM underperforms. However, like SAM, it carries potential risks, such as surveillance overreach, which can raise ethical and privacy concerns.

MM-SAM creates a unified embedding for multiple modalities, which may lead to unintentional associations. Therefore, it is crucial to study joint embedding models, including MM-SAM, carefully to understand these associations and their implications. MM-SAM leverages image embeddings learned by a pretrained model on large web-based data, which can introduce biases, as documented in various studies (Kirillov et al., 2023). For creating unified embeddings for other modalities like thermal, HSI, and LiDAR, we use datasets mentioned in Appendix 6.2. These joint embeddings are limited to the concepts present in these datasets. For instance, the thermal datasets used are limited to outdoor street scenes, while the HSI datasets are confined to remote sensing.

