# OpenReview forum: "Segment Anything with Multiple Modalities"
_ICLR.cc/2025/Conference — Submitted to ICLR 2025_

### Official Review · Reviewer_X7NB · 2024-10-30

**Soundness:** 3
**Presentation:** 3
**Contribution:** 2
**Rating:** 6
**Confidence:** 4

**Summary:**

The paper focuses on extending the foundation model SAM to support other modalities. By inheriting the segmentation priors, MM-SAM achieves remarkable zero-shot performance.

**Strengths:**

1. The MM-SAM is trained in an unsupervised manner with the potential to be easily extended to new modalities and applied to many downstream tasks.

2. MM-SAM supports the multi-modality inference, the segmentation results look good.

3. The method is simple and easy to follow.

**Weaknesses:**

1. Missing training details, please refer to Questions.

2. The UCMT and WMMF lack significant insight, the pseudo-label based training is a typical way in unsupervised and weakly supervised learning.

**Questions:**

1. The SAM demands a prompt input, how to obtain the prompt as input during training? And how about prompt input during inference?

2. If trained MM-SAM is an all-in-one model? Each modality maintains an independent image encoder LoRA?

3. In my understanding, MM-SAM is trained with single or paired modalities, it can be generalized to multi-modality input during inference, is it right? What is the relation with Imagebind? [1]

[1] ImageBind: One Embedding Space To Bind Them All

---

> ### Author Response · Authors · 2024-11-20
> **Reply to Weakness 1 & Question 1**
>
> We sincerely thank you for your valuable feedback, particularly for your recognition of MM-SAM's unsupervised training approach, extensibility, simplicity, strong results, and the clarity of our manuscript. Below, we provide detailed, point-by-point responses to address your concerns.
>
> **[W1\&Q1] More descriptions about prompt input.**
>
> *Response*: Thank you for the comment. We follow [A] and use bounding boxes as input prompts in the training and inference. We will clarify this issue in the updated manuscript.
>
> [A] Segment anything in high quality. NeurIPS 2023.

---

> ### Author Response · Authors · 2024-11-20
> **Reply to Weakness 1 & Question 2**
>
> **[W1\&Q2] If trained MM-SAM is an all-in-one model? Each modality maintains an independent image encoder LoRA?**
>
> *Response*: Thank you for your comment.
>
> MM-SAM is not an all-in-one model but **a flexible framework** (Lines 94-97, 317-318) that can generalize to various multi-sensor combinations, as demonstrated with seven different modalities in both SAM and SAM2 in our experiments. We believe such flexibility enables easy extension to new modalities and diverse downstream tasks, as highlighted by you in the Strengths.
>
> In addition, each non-RGB modality is processed through its own patch embedding module, which handles modality-specific inputs; each modality also has a set of LoRA parameters that capture the unique characteristics of that modality. These parameters are lightweight and computationally efficient, ensuring that the overall model remains resource-efficient, as demonstrated in Table 1 of the manuscript. We will update the manuscript for a clearer description.

---

> ### Author Response · Authors · 2024-11-20
> **Reply to Weakness 2**
>
> **[W2] Clarify the insights of UCMT, WMMF and pseudo labeling.**
>
> *Response*: Thank you for the comment. We clarify that UCMT does not involve pseudo-labeling. Instead, it employs unsupervised feature unification as defined in Equation (1) using paired data.
>
> Regarding pseudo-labeling, we emphasize its necessity and value in MM-SAM. For many sensor modalities and tasks, such as SAR in remote sensing, collecting large annotated datasets is highly impractical due to the significant costs and labor involved. MM-SAM incorporates pseudo-labeling and achieves superior performance without requiring annotations, thereby greatly reducing the burden of data collection and broadening its application scope, as valued by *Reviewer c6Lh* and *you*.
>
> We also highlight that the primary contribution of MM-SAM does not lie in the individual technical modules, but in the **innovative extension and expansion of SAM** toward cross-sensor (single non-RGB modality) and multi-sensor (multiple modalities) segmentation tasks, achieved in a **label-free manner**. However, UCMT and WMMF are carefully tailored to address this new challenge with unique insights and considerations distinct from previous work. These insights, detailed in Lines 302–318, are summarized below:
> - **Novel insight**: SAM’s pre-trained RGB representation space can effectively support non-RGB modalities, which forms the *foundation* of MM-SAM’s technical modules.
>   - **Cross-sensor segmentation**: Building on this shared representation, UCMT enables single-modal segmentation (non-RGB data) through unsupervised cross-modal representation unification for knowledge transfer.
>   - **Multi-sensor fusion**: This unified representation across modalities significantly mitigates the challenges of data heterogeneity. To this end, WMMF learns fusion weights only without altering features of different modalities, enabling superior multi-modal segmentation.
> - **Generalizability**: UCMT and WMMF support both SAM and SAM2 and have been validated across seven different sensor modalities, as shown in Tables 2–5. To our knowledge, few studies on multi-modal segmentation provide such comprehensive evaluations, highlighting the generalizability and robustness of the MM-SAM framework.
>
> MM-SAM represents **a significant paradigm shift** from conventional multi-modal segmentation methods.  It eliminates the need for intensive manual annotations that previous methods need for learning shared representation. We believe this innovation could support a wide range of applications, such as robotics, remote sensing, and autonomous driving, where sensor suites are crucial for robust perception and broader deployment. We will include this discussion in the updated manuscript to highlight the originality and significance of MM-SAM better.

---

> ### Author Response · Authors · 2024-11-20
> **Reply to Question 3**
>
> **[Q3] In my understanding, MM-SAM is trained with single or paired modalities, it can be generalized to multi-modality input during inference, is it right? What is the relation with Imagebind?**
>
> *Response*: Thank you for the comment. MM-SAM is trained with paired modalities to align non-RGB representations with SAM's RGB latent space, enabling both single-modal segmentation and multi-modal fused segmentation.
>
> MM-SAM is a flexible framework capable of generalizing to various combinations of data modality pairs (as shown in Table 1) and extending to three-modality combinations (discussed in Section 4.4.2 and Table 4(b)). However, it requires training the SFG module on paired data for selective fusion, meaning it cannot generalize to arbitrary multi-modal inputs during inference without prior training. Nevertheless, extending MM-SAM to new modalities remains cost-efficient and straightforward as SFG leverages pseudo-labeling without requiring human annotations.
>
> Regarding its relation to ImageBind [1], while MM-SAM and ImageBind share certain broad goal of bridging multiple modalities, their tasks, motivations, and approaches differ significantly:
> - **Different Tasks**: ImageBind is designed for CLIP, focusing on image-level understanding for vision-language modeling. In contrast, MM-SAM is tailored for SAM, addressing pixel-level segmentation tasks (dense predictions) in visual foundation modeling. Due to the task and architectural differences between CLIP and SAM, adapting ImageBind to SAM is not feasible.
> - **Different Insights**: While ImageBind emphasizes universal embedding alignment across modalities for the entire model, MM-SAM introduces a novel insight by demonstrating that cross-modal representation unification can occur within a specific part of the architecture—SAM’s image encoder—while remaining fully compatible with its prompt encoder and mask decoder.
>
> In summary, although both methods involve cross-modal knowledge transfer, MM-SAM is uniquely designed to extend SAM’s capabilities for multi-modal segmentation, addressing challenges specific to dense segmentation tasks and sensor fusion.

---

> ### Author Response · Authors · 2024-11-24
> **Follow-Up on Response Submission and Request for Feedback**
>
> Dear Reviewer X7NB,
>
> We hope this message finds you well. We have submitted a detailed response addressing your valuable feedback and appreciate the opportunity to clarify and strengthen our work.
>
> We kindly request that you review our response and reconsider our submission in light of the clarifications and supporting evidence provided. We are dedicated to ensuring that we fully meet your expectations.
>
> Thank you once again for your time and consideration.
>
> Best regards,
> The Authors

---

> > ### Comment · Reviewer_X7NB · 2024-11-24
> >
> > Thanks for the author's replies, which address parts of my confusion like its flexible design.
> >
> > However, after reading the responses and comments from other reviewers, I have some concerns.
> >
> > (1) All reviewers mentioned the limited novelty, considering the authors place the insight on their novel setting, I maintain neutrality on its novelty contribution.
> >
> > (2) Regarding the ``Reply to Weakness 1 & Question 1'', the pseudo labels are obtained by feeding bounding box prompts to HQ-SAM, which raises concerns about claims of unsupervised learning and fair comparison with previous segmentation methods. Firstly, unsupervised learning seems over-claimed, since the bounding box is not a cheap annotation and is not freely available, the training setting is more closed to weakly supervised learning. Secondly, the inference stage requires bounding box prompts as well, which provide strong priors for segmentation, it is not a fair comparison when compared to previous methods which receive image inputs only, as shown in Table. 10.

---

> > > ### Author Response · Authors · 2024-11-25
> > > **Reply to your concerns**
> > >
> > > **Response to concern (1)**
> > >
> > > Thanks for your feedback. We believe that *novelty in research can emerge not only from technical innovation but also from addressing new, unexplored challenges using effective adaptations of existing techniques*. We hope our novel extension of SAM to multi-sensor segmentation in a mask-free manner serves as an inspiration for future advancements in the field.
> > >
> > > **Response to concern (2)**
> > >
> > > We thank you for your feedback and would like to clarify potential misunderstandings in our manuscript:
> > >
> > > 1. Clarification on Unsupervised Learning:
> > > - We did not claim pseudo-labeling and re-training as fully unsupervised approaches. Instead, they are exclusively part of the WMMF module (**Weakly-supervised Multi-Modal Fusion**, Section 3.2.2), since they involve prompt inputs. This is aligned with your point.
> > > - In contrast, the UCMT module (Unsupervised Cross-Modal Transfer, Section 3.2.1) is fully unsupervised, optimized using the embedding unification function described in Equation 1, without requiring any prompt inputs (Lines 292–296) and human annotations.
> > >
> > > Therefore, we believe our claims are properly justified.
> > >
> > > 2. Comparison with Segmentation-specific Methods:
> > > - Table 10 in the *appendix* is provided **for comprehensive evaluation**. Comparisons between SAM/MM-SAM (promptable segmentation models) and traditional semantic segmentation methods are not strictly fair due to differences in their settings, as we explained in Lines 988–993 of the manuscript. Nevertheless, these comparisons offer valuable insights into the performance of visual foundation models like SAM and MM-SAM on similar tasks (Lines 993–994).
> > > - As recognized by all reviewers, MM-SAM represents the first study to extend SAM from RGB into multi-sensor segmentation. For this reason, SAM serves as the **only strict baseline**, and this comparison is emphasized in the main paper, while Table 10 offers additional contexts and insights in the appendix.
> > > - While MM-SAM uses prompt priors for segmentation as SAM does, it does not rely on mask annotations as strong training supervision like traditional semantic segmentation methods. This demonstrates its superiority from a broader perspective.

---

> > > > ### Comment · Reviewer_X7NB · 2024-11-26
> > > >
> > > > I greatly appreciate the author's replies. I understand that the training combines unsupervised learning and learning under weakly supervised pseudo-labels. Due to the fact that this setting cannot be strictly aligned with previous work, I can not determine its position in segmentation. I look forward to discussing this further with other reviewers before final decision.

---

### Official Review · Reviewer_c6Lh · 2024-11-01

**Soundness:** 3
**Presentation:** 3
**Contribution:** 3
**Rating:** 6
**Confidence:** 3

**Summary:**

MM-SAM expands the capabilities of the SAM to work with multi-modal data from sensor suites, enabling segmentation of data from diverse sensors, both individually and in combination. MM-SAM achieves this by adapting SAM's image encoder, which was originally trained on RGB images, to effectively handle other modalities, such as depth, thermal, LiDAR, and hyperspectral data.

The framework addresses three main challenges: adapting SAM to diverse non-RGB sensors, fusing data from multiple sensors, and enabling mask-free training. To overcome these challenges, MM-SAM introduces two modules: Unsupervised Cross-Modal Transfer (UCMT) and Weakly-supervised Multi-Modal Fusion (WMMF).

UCMT incorporates a modal-specific patch embedding module and parameter-efficient tuning using LoRA within SAM's image encoder. This allows MM-SAM to extract features unique to each sensor modality while keeping the number of added parameters low. A crucial part of UCMT is the embedding unification loss, which aligns the embeddings of different modalities within the output latent space of SAM's image encoder.

WMMF introduces a lightweight selective fusion gate to adaptively fuse multi-modal embeddings. This gate dynamically adjusts the weighting of embeddings from different modalities, effectively using the strengths of each sensor in diverse situations.

Extensive experiments on seven datasets demonstrate MM-SAM's effectiveness.

**Strengths:**

1. Pioneering Work: MM-SAM is the first study to extend visual foundation models, specifically SAM, for use with data from multiple sensor suites.

2. Simplicity and Efficiency: MM-SAM is designed to be technically straightforward, using relatively simple modules like UCMT and WMMF for adaptation. This simplicity makes it easier to understand and implement.

3. Label-Free Adaptation: MM-SAM can adapt to different sensors without requiring mask annotations. It significantly reduces the effort and cost associated with data annotation.

4. Robustness and Versatility: MM-SAM shows superior effectiveness across a wide range of sensor modalities and diverse scene types. This broad applicability makes MM-SAM a valuable tool for various downstream tasks.

5. Shared Latent Space: The output latent space of SAM's RGB image encoder can be effectively shared across different sensor modalities. This shared space allows MM-SAM to generate modality-specific embeddings that are still compatible with the other components of SAM (the prompt encoder and mask decoder), leading to effective cross-modal segmentation. The shared latent space also simplifies multi-modal fusion, allowing for adaptive weighting of embeddings from different sensors.

6. Strong Empirical Results: Extensive experiments across a diverse range of datasets and sensor modalities provide compelling evidence for MM-SAM's effectiveness.

**Weaknesses:**

1. Lack of Novelty in Individual Techniques: While the combination and application of these techniques to adapt SAM for multi-modal data might be novel, the lack of groundbreaking innovation in the individual methods could be seen as a weakness.

2. Limited Comparison with Segmentation-Specific Methods: The experiments primarily compare MM-SAM with SAM and SAM 2, without comparing it to other state-of-the-art methods specifically designed for segmentation tasks, particularly those tailored for specific modalities. For example, on SemanticKITTI, existing LiDAR-specific algorithms can achieve higher mIOU (2DPASS 72.9) on single-mode data without additional training data compared to MM-SAM's performance of 66.4.

**Questions:**

Please refer to the weakness.

---

> ### Author Response · Authors · 2024-11-20
> **Reply to Weakness 1**
>
> We sincerely thank you for your valuable feedback, particularly for your recognition of the novelty, simplicity, versatility, and efficiency of our approach, as well as the effectiveness and comprehensiveness of our experimental results. Please find our responses to your concerns as follows.
>
> **[W1] Clarification of originality and contributions of individual techniques in MM-SAM.**
>
> *Response*: Thank you for your feedback. We would first clarify that our main contribution lies not in the individual technical modules but in the **innovative extension and expansion of SAM** toward cross-sensor (single non-RGB modality) and multi-sensor (multiple modalities) segmentation tasks, achieved in a **label-free manner**, as you recognized.
>
> As detailed in Lines 98-99, MM-SAM is the first study to tackle this challenge, with meticulous technical designs and novel insights that operate under distinct logistics and considerations compared to prior works (Lines 302–318). We summarize its uniqueness as compared with prior studies as follows:
>
> - **Novel insight**: We show that SAM’s pre-trained RGB representation space can be effectively shared across non-RGB modalities, forming the foundation of our technical modules.
>   - **Cross-sensor segmentation**: Building on this insight, we directly implement cross-modal feature unification to transfer knowledge from SAM’s pre-trained RGB latent space to other modalities. A straightforward parameter-efficient tuning in UCMT enables effective cross-modal segmentation without requiring human annotations.
>   - **Multi-sensor fusion**: The unified representations across modalities significantly mitigate the challenges of data heterogeneity. To this end, we design the SFG module, which learns fusion weights only without altering features of different modalities, enabling superior multi-modal segmentation.
> - **Generalizability**: We conducted extensive experiments, as shown in Tables 2–5, to demonstrate MM-SAM’s compatibility with seven different sensor types across both SAM and SAM2 frameworks. To our knowledge, few studies on multi-modal segmentation provide such comprehensive evaluations, highlighting the generalizability and robustness of MM-SAM's framework.
>
>
> MM-SAM represents **a significant paradigm shift** from conventional multi-modal segmentation methods.  It eliminates the need for intensive manual annotations that previous methods need for learning shared representation. Hence, MM-SAM has great potential to support a wide range of applications, such as robotics, remote sensing, and autonomous driving, where sensor suites are crucial for robust perception and broader deployment. We will include this discussion in the updated manuscript to highlight the originality and significance of MM-SAM better.

---

> ### Author Response · Authors · 2024-11-20
> **Reply to Weakness 2**
>
> **[W2] More comparison with segmentation-specific methods.**
>
> *Response*: Thank you for the comment. We clarify that Table 10 in the appendix includes comparisons to state-of-the-art methods designed for segmentation tasks (*sourced from paperwithcode*), using widely adopted datasets such as SUN RGB-D and MFNet. While MM-SAM’s primary goal is to extend SAM for cross-sensor and multi-sensor segmentation, these comparisons provide a broader evaluation of MM-SAM’s performance. We copy the Table 10 below for your quick reference.
>
> | Model              | mIoU on **SUN RGB-D** (RGB+Depth) |
> |--------------------|-----------------------------------|
> | CMX                | 52.4                              |
> | DFormer-L          | 52.5                              |
> | DPLNet             | 52.8                              |
> | TokenFusion        | 53.0                              |
> | GeminiFusion       | 54.6                              |
> | SAM (RGB)          | 78.7                              |
> | SAM (Depth*)       | 68.1                              |
> | MM-SAM (Depth)     | 77.2                              |
> | MM-SAM (RGB+Depth) | 81.2                              |
>
>
> | Model                | mIoU on **MFNet** (RGB+Thermal) |
> |----------------------|---------------------------------|
> | DPLNet               | 59.3                            |
> | CMX                  | 59.7                            |
> | CMNeXt               | 59.9                            |
> | Sigma-base           | 61.3                            |
> | CRM\_RGBTSeg         | 61.4                            |
> | SAM (RGB)	           | 71.5                            |
> | SAM (Thermal*)       | 68.2                            |
> | MM-SAM (Thermal)     | 75.2                            |
> | MM-SAM (RGB+Thermal) | 78.4                            |
>
> It is important to note that the benchmarked methods, including the suggested 2DPASS, achieve high performance by leveraging *full supervision and access to large-scale, densely annotated datasets*. In contrast, MM-SAM is designed for label-free adaptation. Additionally, these approaches focus on semantic segmentation, whereas MM-SAM focuses on promptable segmentation as SAM. These fundamental differences make direct comparisons less equitable.
>
> Nevertheless, MM-SAM demonstrates highly competitive performance despite its unsupervised nature. As shown in the tables, MM-SAM significantly outperforms state-of-the-art supervised methods on SUN RGB-D and MFNet. Even on the suggested SemanticKITTI, MM-SAM achieves a strong result (66.4 vs. 72.9 by 2DPASS), which is particularly noteworthy given that MM-SAM requires no manual annotations as its training ground truth.

---

> ### Author Response · Authors · 2024-11-24
> **Follow-Up on Response Submission and Request for Feedback**
>
> Dear Reviewer c6Lh,
>
> We hope this message finds you well. We have submitted a detailed response addressing your valuable feedback and appreciate the opportunity to clarify and strengthen our work.
>
> We kindly request that you review our response and reconsider our submission in light of the clarifications and supporting evidence provided. We are dedicated to ensuring that we fully meet your expectations.
>
> Thank you once again for your time and consideration.
>
> Best regards,
> The Authors

---

> > ### Comment · Reviewer_c6Lh · 2024-11-26
> >
> > I appreciate the authors' response and will keep my score after reviewing the rebuttal.

---

### Official Review · Reviewer_HTYB · 2024-11-02

**Soundness:** 3
**Presentation:** 3
**Contribution:** 3
**Rating:** 5
**Confidence:** 4

**Summary:**

In this article, the author extends SAM from a single RGB modality to a multimodal approach. To achieve this, they designed three key modules: (1) a fine-tuned image encoder to process X-modal data, aligning its encoded features with those of RGB data and effectively utilizing SAM's pre-trained mask and prompt encoders; (2) a selective layer for efficient fusion of encoded features from different modalities; and (3) the use of multimodal pseudo-labels to optimize network parameters, equipping the model with the ability to extend effectively to multimodal tasks. Experimental results demonstrate the effectiveness of the proposed design.

**Strengths:**

1.This work is the first to extend SAM from single-modal to multi-modal data.

2.The experiments are comprehensive, and the fine-tuned model contributes valuable insights for advancing the field.

3.The modules designed by the author effectively leverage SAM's existing capabilities, and the fine-tuned model demonstrates promising zero-shot potential.

**Weaknesses:**

1.The modules proposed by the authors are derived from prior work, which may limit the originality of this study.

LoRA does not appear to be a novel technology. The low-rank technology is first proposed in 2021:

[1] Hu, Edward J., et al. "Lora: Low-rank adaptation of large language models." arXiv preprint arXiv:2106.09685 (2021).

And, it has been widely used across NLP and CV fields. The related works about combination of SAM and LoRA:

[2] Lu, Xiaoyan, and Qihao Weng. "Multi-LoRA Fine-Tuned Segment Anything Model for Urban Man-Made Object Extraction." IEEE Transactions on Geoscience and Remote Sensing (2024).

[3] Cai, Lingcong, et al. "Towards Cross-Domain Single Blood Cell Image Classification Via Large-Scale Lora-Based Segment Anything Model." 2024 IEEE International Symposium on Biomedical Imaging (ISBI). IEEE, 2024.

The SFG is seems like a adaptive features fusion module and the similar idea has been used in many fields, for examples:

[1] Li, Xiangtai, et al. "Gated fully fusion for semantic segmentation." Proceedings of the AAAI conference on artificial intelligence. Vol. 34. No. 07. 2020.

[2] Li, Yang, et al. "Self-attention enhanced selective gate with entity-aware embedding for distantly supervised relation extraction." Proceedings of the AAAI conference on artificial intelligence. Vol. 34. No. 05. 2020.

[3] Chen, Xiaokang, et al. "Bi-directional cross-modality feature propagation with separation-and-aggregation gate for RGB-D semantic segmentation." European conference on computer vision. Cham: Springer International Publishing, 2020.

Please detail the difference between the propose methods and the above.

2.The impact of the selective layer is unclear due to a lack of ablation studies. Will the results be greatly affected by removing the SFG module?

3.SAM’s performance on uncommon data distributions (e.g., remote sensing and earth observation data) tends to be limited. Relying
solely on pseudo labels without manual annotation could constrain the effectiveness of MM-SAM. Please provide additional experiments or analysis specifically on these uncommon data distributions.

4.The methodology lacks details on how the model fuses single-modal mask predictions \(M_I\) and \(M_X\) from RGB and X-modal data. Please provide the details about fusion algorithm or method description.

**Questions:**

See weakness.

---

> ### Author Response · Authors · 2024-11-20
> **Reply to Weakness 1**
>
> We sincerely thank you for your valuable feedback, particularly your recognition of the novelty of our approach, as well as the effectiveness and comprehensiveness of our experimental results. In the following, we address your concerns in detail.
>
> **[W1] Originality of MM-SAM and difference from suggested studies.**
>
> *Response*: Thank you for the feedback and the shared prior studies.
>
> We would clarify that the primary contribution of MM-SAM does not lie in the individual technical modules but in the **innovative extension and expansion of SAM** from single RGB segmentation toward cross-sensor (single non-RGB modality) and multi-sensor (multiple modalities) segmentation tasks, achieved in a **label-free manner**, as recognized by all reviewers.
>
> By addressing under-explored challenges, the framework of MM-SAM demonstrates significant originality, as detailed in Lines 302–318 of the manuscript. Below please find a brief summary:
> - **Novel insight**: We show that SAM’s pre-trained RGB representation space can be effectively shared across non-RGB modalities, forming the foundation of our technical modules.
>   - **Cross-sensor segmentation**: Building on this insight, we develop a straightforward cross-modal feature unification to transfer knowledge from SAM’s pre-trained RGB latent space to other modalities. We adopt LoRA [1] for parameter-efficient tuning (PEFT) due to its wide usage and lightweight design (Lines 231–232), though other PEFT methods such as Adaptformer and VPT could also be applied, as demonstrated in Fig.5 (a).
>   - **Multi-sensor fusion**: The unified representation across modalities significantly mitigates the challenges of data heterogeneity. To this end, we design the SFG module, which learns fusion weights only without altering features of different modalities, enabling superior multi-modal segmentation.
> - **Generalizability**: Extensive experiments, as shown in Tables 2–5, demonstrate MM-SAM’s compatibility with seven different sensor types across both SAM and SAM2 frameworks. To our knowledge, few studies on multi-modal segmentation provide such comprehensive evaluations, highlighting the generalizability and robustness of MM-SAM's framework.
>
> MM-SAM introduces a paradigm shift in multi-modal segmentation by aligning representations from SAM’s RGB latent space for multiple sensor modalities without human labels. This contrasts with the suggested studies with fundamentally different scopes, objectives and tasks:
> - LoRA-based studies [2, 3] fine-tune SAM *within RGB domains*, including remote sensing [2] and blood cell classification [3], using *fully supervised* frameworks that rely on *large-scale human annotations*.
> - For Fusion modules: [1] uses gated fusion for multi-scale feature integration in *fully supervised RGB* semantic segmentation. [2] applies selective gating for *relation classification in NLP*, which is a fundamentally different domain. [3] introduces SA-Gate to *recalibrate RGB and depth features* in a *fully supervised* way.
>
> We thank you for your suggestions. We will review the suggested studies in the updated manuscript to better illustrate MM-SAM’s originality and its contributions to the field.

---

> ### Author Response · Authors · 2024-11-20
> **Reply to Weakness 2**
>
> **[W2] The ablation of the SFG module.**
>
> *Response*: Thank you for the comment. The SFG module is designed specifically for fusing multi-modal features, and removing it would result in the loss of multi-modal segmentation capabilities.
>
> To examine the effectiveness of SFG as suggested, we conduct an ablation study and compare three different fusion strategies. The math notations are consistent with those in the manuscript:
> - **Max fusion**: The maximum prediction probabilities from RGB and non-RGB feature embeddings are selected at each position: $\hat{e} _ {F _ i} = \max(e_{I_i}, e_{X_i})$. Here, embeddings ($e_I$, $e_X$) are encoded from RGB images and X-modal data $(I, X)$, respectively, with $i$ denoting the patch index.
> - **Avg fusion**: The average of feature embeddings from both modalities is used as the fused embedding: $\hat{e} _ F = \frac{1}{2}(e_{I_i} + e_{X_i})$.
> - **SFG fusion**: The SFG module adaptively learns fusion weights $\omega_i$, as implemented in MM-SAM, following Equation (2): $\hat{e} _ F = \omega_i e_{I_i} + (1 - \omega_i) e_{X_i}$.
>
> We also include SAM’s performance using only RGB inputs for reference. Several points can be concluded from the table below:
> - **Max fusion**: Naively selecting maximum probabilities leads to suboptimal fusion segmentation performance.
> - **Avg fusion**: This fusion strategy can be considered as a **special case** of SFG by assigning equal weights ($\omega_i = 0.5$). This fusion strategy consistently improves segmentation over SAM, confirming that weighted fusion benefits cross-modal integration. This aligns with our core insight: unifying representations in SAM’s RGB latent space enables effective multi-modal integration, even with simple fusion methods.
> - **SFG fusion**: Unlike Avg fusion, which treats all modalities equally and overlooks their complementary strengths, SFG dynamically adjusts to sensor-specific inputs, as shown in Fig. 4. By leveraging the strengths of each modality, SFG delivers the best segmentation robustness and accuracy across all datasets.
>
> The results further validate the significance of our insights and the effectiveness of the SFG design. We thank you for your suggestions and will include this analysis and discussion in the revised manuscript.
>
> | Fusion                    | SUN RGB-D (RGB+depth) | MFNet (RGB+Thermal) | DFC18 (RGB+HSI) | DFC18 (RGB+MS-LiDAR) | DFC23 (RGB+SAR) |
> |---------------------------|-------------------|-----------------|-------------|------------------|-------------|
> | SAM (RGB only w/o fusion) | 78.7              | 68.2/72.6/65.1  | 78.1        | 78.1             | 75.3        |
> | Max (newly-included)      | 76.2              | 73.4/73.7/72.0  | 80.6        | 83.6             | 74.5        |
> | AVG (newly-included)      | 80.2              | 74.7/75.0/73.3  | 85.8        | 86.9             | 77.1        |
> | SFG                       | 81.2              | 75.9/74.7/74.7  | 88.5        | 87.9             | 77.4        |

---

> ### Author Response · Authors · 2024-11-20
> **Reply to Weakness 3**
>
> **[W3] More results or analysis for SAM’s performance on uncommon data distributions such as remote sensing.**
>
> *Response*: We emphasize the use of pseudo-labeling in MM-SAM because collecting large-scale annotated datasets for many sensor modalities, such as SAR in remote sensing, is challenging and resource-intensive. The pseudo-labeling in MM-SAM allows for achieving competitive performance without requiring manual annotations, greatly reducing the cost and effort of data collection and broadening its applicability to diverse domains, as highlighted by *Reviewers c6Lh* and *X7NB*.
>
> On the other hand, MM-SAM is flexible and can operate with annotated data when available. Specifically, pseudo-labels in WMMF’s SFG module can be replaced with ground-truth annotations to train fusion weights. To evaluate this, we conducted experiments on both MM-SAM and MM-SAM2 using ground-truth labels (**“GT”**) and compared them to the original pseudo-label-based results (**“PL”**). The results of four remote sensing benchmarks are summarized in the table below:
>
> - *DFC23 (RGB+SAR) and Potsdam (RGB+DSM)*: Models trained with ground-truth annotations achieved slightly higher performance compared to pseudo-label-based training.
> - *DFC18 (RGB+MS-LiDAR/HSI)*: Models trained with pseudo-labels performed slightly better than those trained with ground truth.
>
> These results demonstrate that pseudo-labels provide comparable supervision to human annotations in fusion learning, highlighting MM-SAM’s superiority in single-modal prediction and SFG’s effectiveness in achieving adaptive fusion. These findings further validate the practicality and robustness of MM-SAM’s design, particularly in scenarios where annotated data is scarce or unavailable. We thank you for your suggestions and will include this analysis and discussion in the revised manuscript.
>
> | Model    | Label               | DFC18 (RGB+HSI) | DFC18 (RGB+MS-LiDAR) | DFC23 (RGB+SAR) | Potsdam (RGB+DSM) |
> |----------|---------------------|-------------|------------------|-------------|---------------|
> | MM-SAM1  | PL                  | 88.5        | 87.9             | 77.4        | 83.6          |
> | MM-SAM1  | GT (newly-included) | 87.0        | 86.7             | 77.6        | 86.0          |
> | MM-SAM2  | PL                  | 90.3        | 91.0             | 79.4        | 87.3          |
> | MM-SAM2  | GT (newly-included) | 88.5        | 89.2             | 79.5        | 89.4          |

---

> ### Author Response · Authors · 2024-11-20
> **Reply to Weakness 4**
>
> **[W4] More fusion details.**
>
> *Response*: Thank you for the suggestion. As described in Lines 264–265, we adopt a confidence-based fusion strategy to combine single-modal mask predictions from RGB ($M_I$) and X-modal data ($M_X$). The confidence of each pixel $(x, y)$ is defined as the maximum absolute value of logits in the respective mask predictions across modalities. Specifically, we compare the confidence scores at corresponding pixel locations in the RGB and X-modal predictions and select the higher one as the fused prediction, formulated as $M_F(x,y) = \max(M_I(x,y), M_X(x,y))$. This approach ensures that the fused prediction leverages the most confident predictions across modalities at each pixel location, effectively utilizing the complementary strengths of different modalities. We will include this explanation in the updated manuscript to enhance clarity.

---

> ### Author Response · Authors · 2024-11-24
> **Follow-Up on Response Submission and Request for Feedback**
>
> Dear Reviewer HTYB,
>
> We hope this message finds you well. We have submitted a detailed response addressing your valuable feedback and appreciate the opportunity to clarify and strengthen our work.
>
> We kindly request that you review our response and reconsider our submission in light of the clarifications and supporting evidence provided. We are dedicated to ensuring that we fully meet your expectations.
>
> Thank you once again for your time and consideration.
>
> Best regards,
> The Authors

---

> ### Comment · Reviewer_HTYB · 2024-11-26
>
> Based on the experiment results and limitations of the paper as well as the answers of the rebuttal, i decide to keep my score.

---

### Meta-Review · Area_Chair_9TfH · 2024-12-20

**Metareview:**

This submission received feedback from three reviewers, all reviewing for the first time. Their scores were borderline, and their confidence levels were not high. One negative reviewer mainly questioned the paper's contribution and innovation, and I agree with this reviewer's opinion. After rebuttal, the reviewers' comments are as follows.

For Reviewer HTYB, the authors' rebuttal improved the completeness and technical details of the paper but did not fully address the reviewers' overall concerns regarding the novelty and impact of MM-SAM. While the reviewers acknowledged the authors' efforts, the contribution of the paper is still considered marginally below the acceptance threshold.

For Reviewer c6Lh, the authors' rebuttal significantly enhanced the technical details and experimental support of the paper, further demonstrating the value of MM-SAM as an innovative label-free multi-modal segmentation framework. The reviewers maintained their assessment of the paper as "marginally above the acceptance threshold."

For Reviewer X7NB, the reviewers noted that the pseudo-label training in MM-SAM relies on bounding box prompts, which aligns more closely with weakly supervised learning rather than fully unsupervised learning. This raised discussions regarding the training methodology and the level of supervision involved. Additionally, the differences in prompt input and supervision levels compared to traditional segmentation methods were highlighted as potential factors affecting the fairness of comparisons, thereby limiting the ability to precisely position MM-SAM's performance.

All in all, my recommendation is to “reject”.

**Additional Comments On Reviewer Discussion:**

This submission received feedback from three reviewers, all of whom were reviewing for the first time. Their scores were borderline, and their confidence levels were not high. One negative reviewer mainly questioned the paper's contribution and innovation, even after the rebuttal.

---

### Decision · Program_Chairs · 2025-01-22

Reject